# A Convolutional Neural Network-Based Connectivity Enhancement Approach for Autism Spectrum Disorder Detection

**DOI:** 10.3390/jimaging9060110

**Published:** 2023-05-31

**Authors:** Fatima Zahra Benabdallah, Ahmed Drissi El Maliani, Dounia Lotfi, Mohammed El Hassouni

**Affiliations:** 1Laboratory of Research in Information Technology and Telecommunication (LRIT), Rabat IT Center, Faculty of Sciences, Mohammed V University in Rabat, Rabat B.P. 1014 RP, Morocco; a.elmaliani@um5r.ac.ma (A.D.E.M.); d.lotfi@um5r.ac.ma (D.L.); 2Laboratory of Research in Information Technology and Telecommunication (LRIT), Rabat IT Center, lFLSH, Mohammed V University in Rabat, Rabat B.P. 1014 RP, Morocco; mohamed.elhassouni@flsh.um5.ac.ma

**Keywords:** autism, deep learning, Rs-fMRI, connectivity

## Abstract

Autism spectrum disorder (ASD) represents an ongoing obstacle facing many researchers to achieving early diagnosis with high accuracy. To advance developments in ASD detection, the corroboration of findings presented in the existing body of autism-based literature is of high importance. Previous works put forward theories of under- and over-connectivity deficits in the autistic brain. An elimination approach based on methods that are theoretically comparable to the aforementioned theories proved the existence of these deficits. Therefore, in this paper, we propose a framework that takes into account the properties of under- and over-connectivity in the autistic brain using an enhancement approach coupled with deep learning through convolutional neural networks (CNN). In this approach, image-alike connectivity matrices are created, and then connections related to connectivity alterations are enhanced. The overall objective is the facilitation of early diagnosis of this disorder. After conducting tests using information from the large multi-site Autism Brain Imaging Data Exchange (ABIDE I) dataset, the results show that this approach provides an accurate prediction value reaching up to 96%.

## 1. Introduction

Autism spectrum disorder is a neurodevelopmental disorder that has received significant attention over the past five decades. It is among one of the neuropsychiatric syndromes that affects children in the early years of childhood and continues throughout their lives [1], impacting their development, language and social interactions.

ASD exhibits a range of genetic, pathophysiological, and environmental conditions [2]. It is associated with qualitative impairments in social interaction such as problems in using non-verbal behaviors (facial expressions and body postures), failure to develop peer relationships, lack of spontaneous seeking of others, and lack of emotional reciprocity.

Symptoms of ASD also include qualitative impairments in communication as difficulty in language development, inability to initiate or sustain a conversation with others, and stereotyped or idiosyncratic language.

Moreover, autistic children manifest restricted behavior patterns, such as highly focused interests, inflexible adherence to routines, repetitive motor mannerisms, and persistent preoccupation with parts of objects [3]. Often, ASD co-occurs with intellectual disability and other mental disorders including anxiety, depression, aggressive behaviors, repetitive behaviors, inattention with hyperactivity, and sleep disorders [2,4].

Before the age of three years, many abnormalities in social interaction, language, symbolic or imaginative play can be remarked. However, due to the lack of parental perception of such symptoms and the cost or unavailability of diagnostic centers, the average age of autism detection has reached five years old [5]. In addition, the significant heterogeneity of symptoms that co-occur with autism, makes it very challenging to comprehend and diagnose [6]. Therefore, researchers are eager to propose new methods to assist doctors in the early diagnosis of autism and thus to improve the livelihood of autistic children.

Investigation via brain imaging can present a good alternative to the classic behavioral methods. However, structural images can not be completely trusted as the development rate of children in the first years of childhood can vary widely. According to Hussain [7], every child development process is characteristically unique. Therefore, all children do not reach the same point of development at the same age. On the other hand, functional resting-state networks start to emerge before birth and become evident at 26 weeks prenatal age [8]. These networks capture neural interactions and have enabled the detection of autism with an accuracy between 60 and 70% in a heterogeneous setting using machine learning methods [9,10,11]. With the application of deep learning, the accuracy of detection has been shown to further improve reaching 80% [12,13].

Through the achieved results, Epalle et al. [12] concluded that discordance in resting-state network connectivity constitutes a major discriminatory feature between patients with autism and healthy individuals. On the other hand, Kashef [13] declared that there is an anti-correlation of brain function between anterior and posterior areas of the brain. In a previous work, using methods that could be associated with autism deficits as well as a mathematical strategy of elimination, evidence of long range under-connectivity deficit was reported [9].

In this paper, we propose a two-phase system that processes resting-state functional magnetic resonance images (rs-fMRI) [14] of autistic and non-autistic subjects with the objective of correctly detecting autism. The first phase uses fMRI images to extract connectivity maps that represent interactions between different parts of the brain. However, since the brain is composed of billions of small units (synapses), atlases are used to reduce the complexity of the maps as explained in Section 2.

In the second phase, deep learning is introduced to classify the connectivity matrices with a 10-fold cross-validation. Usually, deep learning models present a huge drawback related to the size of data since they require a large number of samples for training. This problem can be solved by data augmentation, but at the cost of the processing time. The use of transfer learning can be a better alternative. According to Kaya et al. [15], transfer learning provides better outcomes when compared with end-to-end models. It also solves the problem of the training phase by using the weights of models that have been trained on known databases such as imageNet [16]. Transfer learning was largely used in the context of brain anomaly detection as in the work of Talo et al., where it was used to detect brain cancer [17]. It was also used in the same frame as our paper to detect autism based on connectivity matrices [18].

The originality of the present paper stems from using a new approach of enhancement based on a specific layering of the data of the CNN network. The proposed layering is ASD tailored since it is inspired by previous autism theories. It highlights connectivity patterns that were proved by Benabdellah et al. [9,10] to contain relevant autism biomarkers, by feeding the CNN with layers representing the properties of over-connectivity and under-connectivity. This paper also puts forward the importance of linking autism findings together with new strategies to achieve better detection. The accuracy of 96% that was reached proves the effectiveness of the proposed approach, especially when considering the heterogeneity of the ABIDE database.

## 2. Material and Methods

### 2.1. Data

The base data of this work are resting-state functional magnetic resonance images (rs-fMRI) proposed by ABIDE I [19]. The latter is a dataset of the Autism Brain Imaging Data Exchange, a base dedicated to autism, that regroups brain images from different sites around the world. Its main objective is to increase the availability of data to forward autism research. In this paper, we use a resting-state context, where the correlation is computed between functionally related brain regions in the absence of any stimulus or task [20]. From the 1112 rs-fMRI images considered, we only select 871 images that were proven free of damage according to Abraham et al. [11]. The actual used data is a component representing the time series of the rs-fMRI images by the parcellation of specific atlases. These parcels, called regions of interest (ROIs), permit reducing the complexity of analysis. They also enhance the comprehensibility of the results, since the synapses of every region have a criterion of similarity that renders comparing their interactions a lot easier. This is supported by Zafar et al. [21] where they stated that parcellation lessens the complexity of the brain analysis and helps in giving more meaning to the results as the clustering is based on a common point that brings the synapses of the brain together into regions.

However, the choice of the parcellation that leads to the best understanding of the brain is still unclear. Therefore, we use atlases that propose different numbers of regions of interest without exceeding a maximum of 200 ROIs. The time series are available through ABIDE preprocessed [22]. As the name suggests, the data is preprocessed using different pipelines and strategies. In this paper, we extract the data preprocessed with C-PAC, which is the most used pre-processing pipeline in the state of the art. We also use the time series of the AAL, DosenBatch and CC200 atlases.

### 2.2. Proposed Approach

The proposed approach involves two important steps. In the first step, we mimic RGB images by constructing three-dimensional connectivity maps that contain different interactions of the brain regions. This step includes a novel enhancement method highlighting the connectivity information crucial to autism detection.

In the second step, we feed the constructed 3D maps to different CNN models in order to predict the existence of autism using a transfer learning approach.

#### 2.2.1. Creation of the 3D Connectivity Matrices

To create the 3D (RGB images alike) connectivity matrices, we first compute time series for every subject using three atlases, namely AAL, DosenBatch, and CC200 atlases. These latter offer several and various sets of clusters of ROIs and permit the construction of connectivity matrices of 116 × 116, 161 × 161, and 200 × 200 connections, respectively.

The connectivity matrices are then computed using three connectivity likelihood methods:The correlation method: calculates the likelihood of communication between ROIs signals. This is done by comparing the activity detected from the time series and allocating weight values between −1 and 1 that represent the strength of the connection between these regions. With the value of 1 as highly correlated.The covariance method [23]: computes coefficients that reflect direct and indirect connections between every two regions. It gives the covariance between each pair of elements as well as the variances that reflect the covariance of each element with itself.The tangent space embedding [24]: permits to go one step further at a group level. It couples the information from all interactions in a unique group connectome using a geometrical framework. Hence, allowing to measure interactions in a common space called the tangent space [25].

From the coupling of the three time series and the connectivity computing methods, nine different connectivity matrices (CM) result for every subject. Each CM is then separately divided by 3 to extract a third of its connectivity weights. The 3D connectivity matrix of this CM is then achieved by transposing the extracted thirds as represented on the right side of Figure 1.

#### 2.2.2. Enhanced 3D Matrices

The enhanced matrices present specific highlighted connections for the layers. They are constructed using the same above dividing process but with information more related to the autism theories. To this end, each connectivity matrix is first transformed into a graph g=(V,E) where V is a finite set of vertices that represents the ROIs of the atlases, and E⊆V×V is a finite set of weighted edges that are the connections between every two ROIs.

Then, the enhanced matrices are extracted based on high-weight and low-weight connections using the maximum spanning tree and the minimum spanning tree, respectively, as shown on the left side of Figure 1. Therefore, the enhanced matrices represent the theories of over-connectivity and under-connectivity, already proven to exist between the brain regions [9,10].

A spanning tree is a subgraph that includes all the vertices of the graph it is applied to and connects them without cycles. Both maximum and minimum spanning trees are extracted using the Kruskal algorithm [26,27]:The Minimum spanning tree (MST): the Kruskal algorithm permits to extract the MST using a greedy approach that selects the lowest weight edge that does not cause a cycle in the MST. The algorithm sorts the edges before constructing the tree by adding increasing arcs at each step, keeping the total weight of all the edges to the minimum.The maximum spanning tree (MaxST): uses the same reasoning but extracts connections that present the highest weights.

Figure 2 visualizes the steps followed to extract MST and MaxSt from the connectivity matrices.

Once both trees are extracted, we construct the enhanced 3D matrices for every subject by interposing the original connectivity matrix on the first layer, the maximum spanning tree on the second layer, and the minimum spanning tree on the third layer (Figure 1).

#### 2.2.3. Classification with CNN

In the second stage, the 3D matrices are fed into CNN models. The convolutional neural network is a deep neural network originally designed for image analysis. It is based on two important operations, namely convolution and pooling. In convolution, multiple filters extract features from the entry data. Then, the pooling operation, also called sub-sampling, is used to reduce the dimensionality of the extracted features. As mentioned before, most models use transfer learning to solve the problem of the small size of the data. To this end, Keras [28], an open-source neural-network library written in Python, provides many trained CNNs with different combinations of convolution and pooling layers. It can run on top of different machine-learning libraries, such as TensorFlow and Theano.

In this work, we use TensorFlow and train seven different models from Keras, namely, ResNet152V2, Inception, ResNet50, InceptionResNet, Xception, VGG19 and VGG16, with the weights of imageNet. Since we are using transfer Learning, the process stops before the fully connected layer, in order to extract features. Then, on top of the last layer, we construct a simple model that takes the previous model’s extracted features as entries. The new model contains a dropout layer whose role is to prevent over-fitting. Then, we apply fine-tuning to improve the classification results.

The main steps of this classification are represented in Figure 3.

In more detail, we use a typical transfer learning workflow, where we instantiate a base model and load pre-trained weights into it. Then, we freeze all layers in the base model to keep the weights as they are in the training process. Afterward, we create a new model on top of the output of the base model. Once the model converges on the extracted features, we unfreeze all the layers of the base model and re-train the whole model end-to-end with a very low learning rate. The objective of this last step is to potentially improve the results.

Moreover, to enhance the reliability of the results, we use a 10-fold cross-validation classification. We conduct ten tests by reserving, every time, one fold for test and keeping the others for training and validation.

Then, we evaluate the classification performance using the accuracy metric that measures the ability of a test to differentiate ASD and control subjects correctly [29] and represent the mean result of all the ten tests of the cross-validation.

The accuracy is defined as follows:(1)$$Accuracy=\frac{TP+TN}{TP+TN+FP+FN}$$

With:TP (True positive) is the number of cases correctly identified as autistics.FP (False positive) is the number of cases incorrectly identified as autistics.TN (True negative) is the number of cases correctly identified as controls.FN (False negative) is the number of cases incorrectly identified as controls.

## 3. Results

Combining the time series of the three atlases with the three connectivity methods produces nine different connectivity matrices for every subject. The number doubles after applying the 3D creation, which results in nine basic 3D matrices and nine enhanced 3D matrices. Hence, for every subject, we end up with 18 different connectivity matrices to analyze.

Feeding these matrices separately to the seven deep learning models results in 128 groups of features. The extracted features are of sizes that vary from 2 × 2 to 7 × 7 depending on the depth of the model and the used atlas.

Then, for every atlas, by using a 10-fold cross-validation we ended up with 42 groups of 10 accuracy values. The mean of these accuracy values are displayed in the tables below. First, Table 1 contains the classification results of the AAL atlas.

Matrices of the AAL atlas have a format of (116, 116, 3). Once fed to the model, these later result in feature sizes of 2 × 2 for Inception and InceptionResNet models, 3 × 3 for the VGG models, and 4 × 4 for the rest.

The first notable point of the results from Table 1 is the number of accuracy values reaching over 90% which are highlighted in red font and tagged with a star. Most of the values presented in this table exceed the state-of-the-art achievement in the case of fMRI-based autism detection. Moreover, the use of the enhancement approach lead to better classification results. Classification of the enhanced 3D matrices permitted to achieve accuracy values of 96.10% and 95.98% using correlation matrices with VGG16 and VGG19, respectively. For the other models, the enhancement approach permitted to reach high values with all the connectivity calculation methods (correlation, covariance, and tangent) except for ResNet50 which achieved high accuracy only with the covariance method. However, VGG16 and VGG19 remain the best choices for modeling the AAL atlas enhanced matrices in general, with all the accuracy values reaching over 90%.

Regarding the DosenBatch atlas and its 161 × 161 × 3 matrices, features are of sizes that vary between 3 × 3 for the Inception and InceptionResNet models, 5 × 5 for the VGG models, and Xception and 6 × 6 for the ResNet Models.

However, when compared to the AAL atlas, the results of Table 2 generally show low values. This implies poor performance of the deep learning models with the features of this atlas. The VGG models that performed their best with the AAL atlas features become the worst with all the strategies and all types of connectivity calculations. Although the enhanced matrices were better classified, the only models to achieve an accuracy of over 90% were Inception and ResNet152, and only with the correlation and tangent methods, respectively.

Since we work with connectivity features, if the deep learning model can not extract decisive biomarkers to differentiate between autistics and non-autistics, the classification will fail. Hence, these poor results could be due to the increase in number of ROIs, and can also be related to the delimitation of the ROIs which may have lead to loss of decisive information.

The CC200 classification results of Table 3 do not align with the theory that an increased number of ROIs leads to poor accuracy results, since many accuracy values exceeded 90%. Although it did not reach the results achieved with the AAL atlas, it still ranked second position. The matrices are of size 200 × 200 × 3. It is worth noting that, in this case, feature sizes vary between 4 × 4 for Inception and InceptionResNet models, 6 × 6 for VGG16 and VGG19 models, and 7 × 7 for all other models. In this Table 3, Xception leads by the highest accuracy of 95.52%, followed by VGG16 with 95.30%. Here, again, the enhancement approached veered the results into high values. However, the overall results were lower than with the AAL atlas. Furthermore, the size of the CC200 matrices makes the execution time the highest among all atlases. As the features sizes are the largest when compared to the other atlases, they induce an extended time of analysis. Hence, the AAL atlas using the enhanced 3D correlation matrices remains the best combination, considering accuracy and time.

After comparing the results of the three tables, we remark that ResNet50 failed for all the correlation and tangent matrices classification and only showed improvement with the covariance. On the other hand, ResNet152 was consistent with the classification of the tangent matrices of the three atlases decreasing by only a minute amount going from AAL to CC200. Inception was also somewhat consistent when considering the atlases but only when considered with the correlation method instead. However, the decrease in accuracy was more notable again going from the AAL atlas to CC200. These results showcase that the structure of the deep learning model is also of great importance. However, one structure can not work for all atlases and all connectivity calculation methods. Finally, Figure 4 shows the results of classification before and after using fine-tuning. It can be clearly seen that this later improves the classification accuracy. We can also remark that the enhancement technique positively impacts the results even without fine-tuning, which supports the results of Table 1, Table 2 and Table 3.

## 4. Discussion

The main goal of the classification task is to categorize the tested subjects into a specific number of defined groups. However, it can also decide the efficiency of the approach used in separating these subjects. Hence, in this paper, we used classification with the objective of testing the efficiency of the enhancement approach in detecting autism. This approach was based on two important steps. The first one consisted of preparing the data and creating 3D matrices that are image alike. The second step was the deep-learning-based classification of the newly created data.

For the sake of comparison with the state-of-the-art, we used ABIDE I, since its pre-processed version was vastly used in many recent works all concerned with early ASD detection [9,30]. Moreover, we have also used this same database with other strategies in previous works to detect autism [9,10]. We also tested different options for creating the base data, creating nine combinations based on three atlases and three connectivity computing methods. Then, we built 3D matrices using two strategies. The first was to keep the information intact by dividing it by three and feeding it to a CNN model. The second strategy introduced enhancement as a technique that considers the deficiency of autism connectivity and enhances it to help deep learning models differentiate autistic subjects from non-autistic subjects.

Using three atlases and three connectivity calculation methods with two 3D creations and seven CNN models with a mix and match approach, resulted in 126 classification strategies. Thus, 126 different accuracy values. Half of them (63) concerned the enhancement strategy. From these, two-thirds were above 70% accurate. Furthermore, by transposing a model with a dropout layer over the transfer learning models and using cross-validation, we increased the reliability of the results and eliminated over-fitting. Then by applying fine-tuning, we increased the accuracy of detection as Figure 4 shows.

Transfer learning permitted the use of deep learning models even with a dataset of only 871 subjects. This made it possible to exploit the benefits other fields could attain using deep learning, as the results show. Note that without deep learning, the best result achieved in the literature of autism detection attained only 70% [9,10,11,24] with the whole ABIDE I dataset. However, in the present paper, we were able to achieve 90% accuracy using the RGB-mimicking to build the 3D matrices. Adding the enhancement strategy, our results leaped to 96% as the highest accuracy.

Hence, from the results achieved with all combinations, we can say that the enhancement strategy was very helpful in increasing the accuracy of detection for most models. This is especially the case regarding the VGG models when coupled with all the connectivity matrices of the AAL atlas, including the correlation, the covariance and the tangent. It is also important to point out that, thanks to the structure of the VGG models that do not comprehend a lot of layers and the small number of ROIs of the AAL atlas, the execution time was much faster compared to the other combinations.

Another interesting point is that there is no correlation between the number of ROIs in an atlas and the classification accuracy. The CC200 (with the highest number of ROIs) and the AAL atlas (with the smallest number of ROIs) permitted the extraction of high-accuracy values with most deep-learning models. However, the Dosenbatch model that comes in between when considering ROIs number did not achieve excellent results. This might be due to the parcellation and the choice of the areas composing the ROIs, since when studying the communication between the regions of the brain, a minimum change in the border of these regions can lead to new findings or to skipping some decisive information. The results also showed that ResNet152V2 permitted accuracy values of over 91% with the tangent enhanced matrices with all atlases. Moreover, ResNet152V2 was the only model that achieved good accuracy values in almost all the scenarios of the enhancement strategy.

Overall, the tangent gave good results in more than half of the models with all atlases except for the DosenBatch, which provided good results only with three of the seven models. Considering the correlation, over 70% of the results have been accurate (above 80% of accuracy), especially with the AAL atlas, where the accuracy exceeded 90%.

Furthermore, the results of Figure 4 show that combining correlation and AAL atlas achieves high-accuracy values even before fine-tuning. However, this combination improves performances in most cases. From the various results achieved, we could also conclude that there is no perfect model or method. Only by testing multiple combinations can we extract the best combination to achieve better autism classification. In the present work, we attained 96.10%, an accuracy never achieved before using the ABIDE dataset. This dataset, known to be heterogeneous, makes it very difficult to detect anomalies accurately.

However, in our work, the heterogeneity became an advantage that permitted the creation of a detection system independent of age that could single out characteristics common to the autistic population.

Furthermore, by focusing on functional rather than structural images, we limited doubts regarding the growth pace difference known to induce different volume sizes of the children’s brains. Moreover, it was reported that the size abnormality of autistic children disappears, and the brain returns to a normal size when reaching adulthood [31]. Therefore, investigating the functional abnormality is more decisive and adapted to any age group.

Furthermore, the use of a large database allowed us to avoid non-generalization of the results usually associated with small data.

Our approach permitted enhancement of the connections likely to be deficient (over- or under-connected) through the creation of 3D matrices in the RGB-mimicking step, which produced high-accuracy values. This shows the importance of using previous findings along with further investigation to achieve early diagnosis of this life-bound disorder and reducing its impact, in the wait for a definitive cure.

## 5. Conclusions

In this paper, we proposed an enhancement strategy that improves autism fMRI-based detection. The approach leverages the previous findings related to connectivity deficiencies in autistic brains through CNN models designed using an RGB-mimicking of over and connectivity matrices. The heterogeneous setting provided by ABIDE with data from different locations and with different subject’s ages, IQ scores, and other criteria help generalize the findings. We succeeded in reaching an accuracy of 96.10%, a very important value demonstrating that we are on the right path to achieving early diagnosis in the near future.

## Figures and Tables

**Figure 1 jimaging-09-00110-f001:**
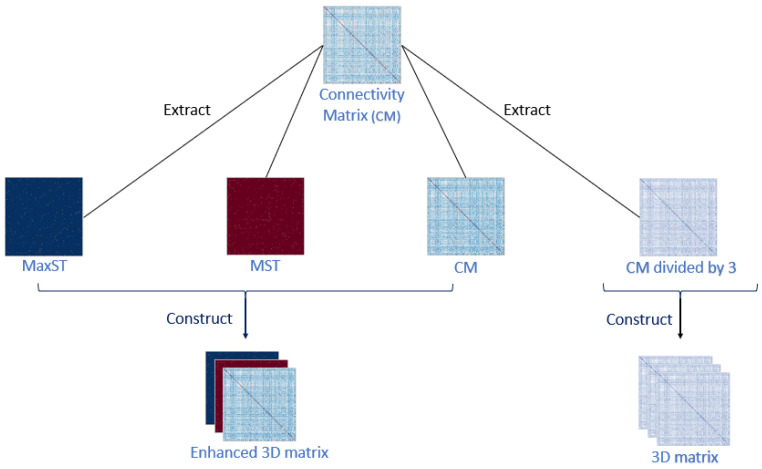
The 3D matrices construction process. MST and MaxST represent the minimum spanning tree and the maximum spanning tree, respectively.

**Figure 2 jimaging-09-00110-f002:**
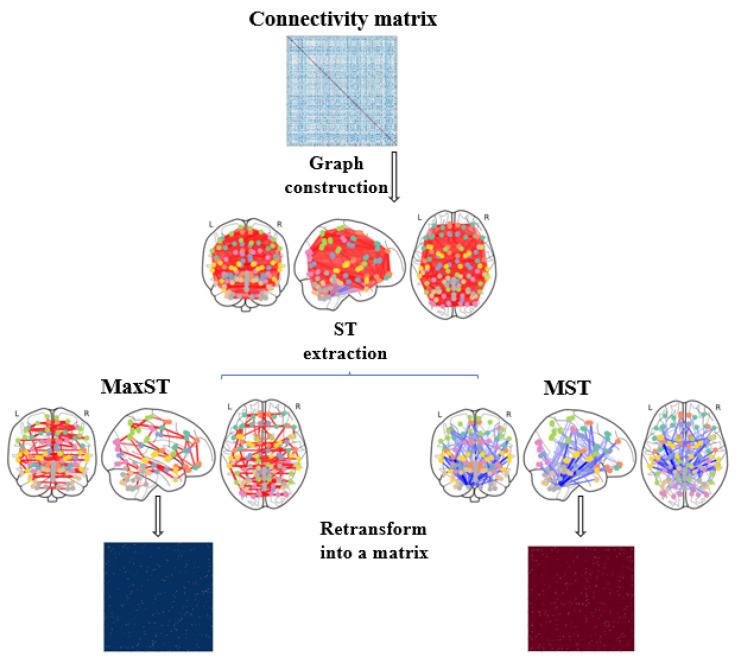
Steps to extract the Spanning Trees matrices from a connectivity matrix.

**Figure 3 jimaging-09-00110-f003:**
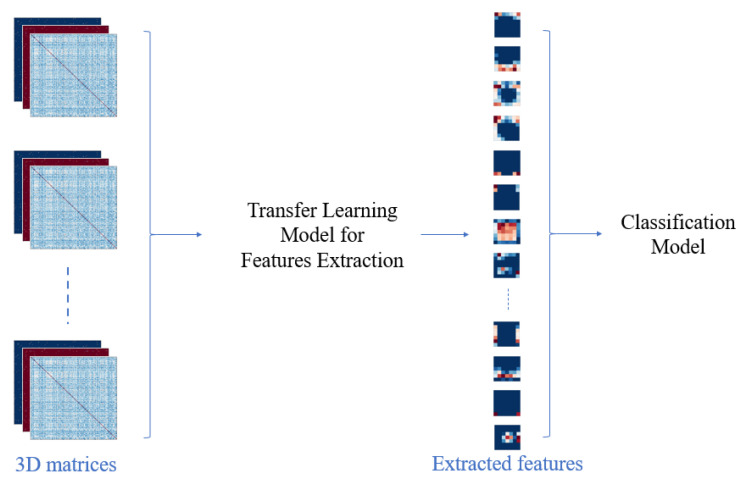
The main steps of deep learning classification.

**Figure 4 jimaging-09-00110-f004:**
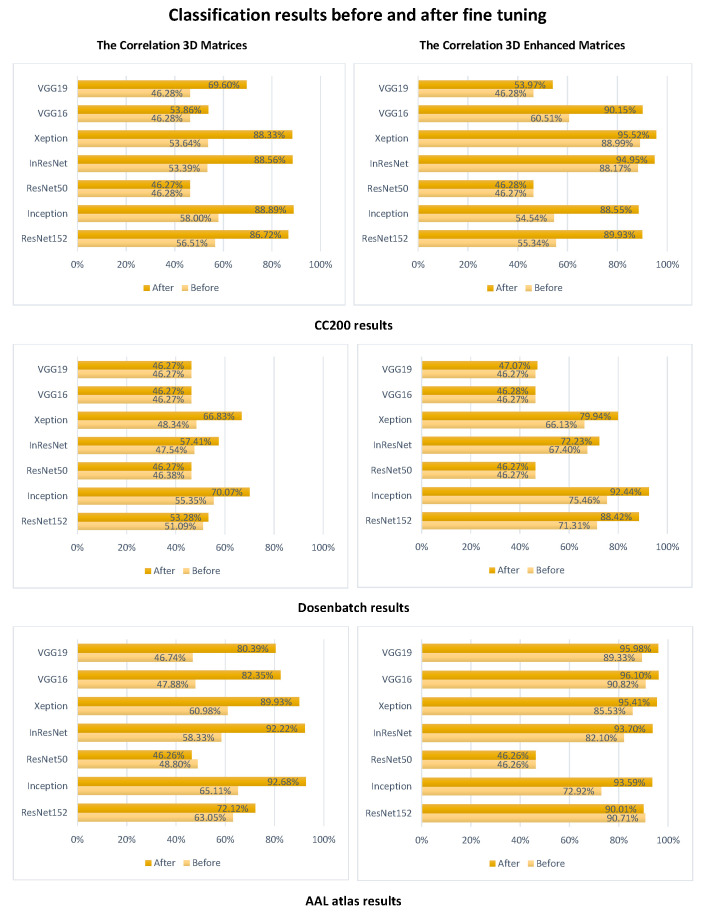
Classification before and after tuning.

**Table 1 jimaging-09-00110-t001:** Accuracy of the classification of the AAL 3D matrices with and without enhancement. Values over 90% are highlighted with red and ’*’.

	Correlation		Covariance		Tangent	
	3D Matrices	Enhanced 3D Matrices	3D Matrices	Enhanced 3D Matrices	3D Matrices	Enhanced 3D Matrices
ResNet152	72.12%	90.01% *	76.14%	87.97%	87.77%	91.85% *
Inception	92.68% *	93.59% *	82.81%	85.80%	84.99%	94.39% *
ResNet50	46.26%	46.26%	85.44%	93.69% *	46.26%	46.27%
InResNet	92.22% *	93.70% *	47.75%	46.85%	85.58%	92.68% *
Xeption	89.93%	95.41% *	64.42%	78.09%	90.04% *	93.58% *
VGG16	82.35%	96.10% *	73.84%	91.51% *	61.91%	92.89% *
VGG19	80.39%	95.98% *	70.75%	90.48% *	68.57%	94.04% *

**Table 2 jimaging-09-00110-t002:** Accuracy of the classification of the DosenBatch 3D matrices with and without enhancement. Values over 90% are highlighted with red and ’*’.

	Correlation		Covariance		Tangent	
	3D Matrices	Enhanced 3D Matrices	3D Matrices	Enhanced 3D Matrices	3D Matrices	Enhanced 3D Matrices
ResNet152	53.28%	88.42%	58.67%	66.95%	65.12%	91.63% *
Inception	70.07%	92.44% *	60.16%	64.08%	49.94%	48.22%
ResNet50	46.27%	46.27%	60.41%	77.98%	46.27%	46.27%
InResNet	57.41%	72.23%	49.25%	47.42%	60.41%	81.20%
Xeption	66.83%	79.94%	60.97%	67.18%	63.85%	76.82%
VGG16	46.27%	46.28%	50.07%	59.24%	46.27%	46.27%
VGG19	46.27%	47.07%	57.98%	58.80%	46.26%	46.26%

**Table 3 jimaging-09-00110-t003:** Accuracy of the classification of the CC200 3D matrices with and without enhancement. Values over 90% are highlighted with red and ’*’.

	Correlation		Covariance		Tangent	
	3D Matrices	Enhanced 3D Matrices	3D Matrices	Enhanced 3D Matrices	3D Matrices	Enhanced 3D Matrices
ResNet152	86.72%	89.93%	61.56%	89.23%	78.90%	91.52% *
Inception	88.89%	88.55%	72.34%	87.39%	87.40%	92.78% *
ResNet50	46.27%	46.28%	84.99%	94.38% *	46.28%	46.27%
InResNet	88.56%	94.95% *	49.96%	52.93%	84.78%	93.70% *
Xeption	88.33%	95.52% *	73.27%	75.21%	87.86%	93.01% *
VGG16	53.86%	90.15% *	93.68% *	95.30% *	46.26%	51.67%
VGG19	69.60%	53.97%	73.85%	90.13% *	46.04%	46.27%

## Data Availability

The original data used in this work are available through https://fcon_1000.projects.nitrc.org/indi/abide/abide_I.html (accessed on 1 January 2022) and the pre-processed version at http://preprocessed-connectomes-project.org/abide/ (accessed on 1 January 2022).

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
