# Peer review of "A Convolutional Neural Network-Based Connectivity Enhancement Approach for Autism Spectrum Disorder Detection"

_2313-433X, 2023, doi:10.3390/jimaging9060110_

Round 1
Reviewer 1 Report
Dear authors,
Your study addresses the important issue of reliable early detection of autism and autism spectrum disorders, using non-invasive imaging and deep learning techniques. While I completely agree that this is an important issue that we definitely must improve on, there are a couple of issues that need to be taken care of before publishing.
Overall, the use of the English language and specific expressions is unsatisfactory, sometimes even completely inappropriate (for example the expression "touched children" should be replaced with "affected children)
Although you are referring to early detection in children from the beginning, the databases you use have a wide range of ages within their subjects (data in ABIDE II represents images from patients with an age range of 5-64 years). This issue must be clarified and it has to be specified how you control for age-related differences and its extrapolation. Evenmore, you mention in the text that: "structural images cannot be completely trusted as the development rate of children in the first years of childhood can be widely different". How do you integrate these caveats in the final interpretation of the results is not clarified.
In the conclusions you make the claim: „enhancement strategy that improves autism detection”. While I agree that you demonstrated clearly the advantages of your method, I think this claim overstates things. I would suggest changing it to something like "improves detection of imaging characteristics usually associated with autism".
Also, you presented different strategies, which expectedly yielded slightly different results. A comparison between the efficacies of the different methods and conditions and an emphasis on the best one would help follow the results and proposed methods.
Last, but not least, there is a weird way of introducing citations: "This is supported by [20]”. I strongly believe that you should use more common way of citing the first author et.al and use the reference number at the end of the phrase.
The same goes for abbreviations. They have to be introduced in the body of the text when you first mention them. The most important one for the paper CNN, although present in the abbreviation list, it is not explained in the body of the text.
Please see the comments and suggestions for English language quality assessments.
Reviewer 2 Report
The article proposes a framework to achieve early and accurate diagnosis of autism spectrum disorder. Previous studies have suggested theories of under and over-connectivity deficits in the autistic brain, and this framework takes these properties into account using an approach of enhancement coupled with deep learning. The approach involves creating image-like connectivity matrices and enhancing connections related to connectivity alterations. The tested dataset is the Autism Brain Imaging Data Exchange, and the results show that this approach provides accurate predictions up to 96%.
I would specify in the title what is CNN
CNN is not mentioned in the abstract
PDD is a term used in the DSM-IV … it is not commonly used anymore
Figures 1 is not very easy to read, can the text and the images be made bigger?
Tables are not appropriate for color-blind person. Can you add and * for the >90%?
NA
Reviewer 3 Report
No title and header for the first section, which is a mixture of introduction and methods.
Bad editing and use of the indent.
Results presentation is very little and can be improved
Discussion section is fine
English language can be improved
